# Elucidating a locus coeruleus-dentate gyrus dopamine pathway for operant reinforcement

**Elijah A Petter[1], Isabella P Fallon[1], Ryan N Hughes[1], Glenn DR Watson[1], Warren H Meck[1], Francesco Paolo Ulloa Severino[1,2], Henry H Yin[1,3]\***

[1]Department of Psychology and Neuroscience, Duke University, Durham, United States; [2]Department of Cell Biology, Duke University School of Medicine, Durham, United States; [3]Department of Neurobiology, Duke University School of Medicine, Durham, United States

**Abstract** Animals can learn to repeat behaviors to earn desired rewards, a process commonly known as reinforcement learning. While previous work has implicated the ascending dopaminergic projections to the basal ganglia in reinforcement learning, little is known about the role of the hippocampus. Here, we report that a specific population of hippocampal neurons and their dopaminergic innervation contribute to operant self-stimulation. These neurons are located in the dentate gyrus, receive dopaminergic projections from the locus coeruleus, and express D1 dopamine receptors. Activation of D1 + dentate neurons is sufficient for self-stimulation: mice will press a lever to earn optogenetic activation of these neurons. A similar effect is also observed with selective activation of the locus coeruleus projections to the dentate gyrus, and blocked by D1 receptor antagonism. Calcium imaging of D1 + dentate neurons revealed significant activity at the time of action selection, but not during passive reward delivery. These results reveal the role of dopaminergic innervation of the dentate gyrus in supporting operant reinforcement.

**\*For correspondence:**
hy43@duke.edu

**Competing interest:** The authors declare that no competing interests exist.

## Editor's evaluation

These important findings indicate that dopamine signaling, arising from the locus coeruleus, and D1R expressing neurons in the dentate gyrus can support positive reinforcement. This is an exciting finding given the prior dearth of information on the role of dopamine signaling in the dentate gyrus. The evidence to support the claims is compelling. Rigorous optogenetic experiments, site specific pharmacology, tracing, and calcium imaging bring together a compelling argument that dopamine signaling in the dentate can play an important role in positive reinforcement. This manuscript will be of interest to those interested in dopamine, locus coeruleus and/or hippocampal function, learning or motivated behaviors.

## Introduction

In operant learning, animals modify their action repertoires to earn desired rewards. Previous work on the neural substrates of such learning has focused on the striatum and the midbrain dopaminergic projections that target the striatum (*Wise, 2004*; *Yin et al., 2005*; *Kravitz et al., 2013*; *Rossi et al., 2013*; *Yttri and Dudman, 2016*). Midbrain dopamine neurons have been implicated in reinforcement learning (*Schultz et al., 1997*; *Tsai et al., 2009*; *Rossi et al., 2013*). Such learning is often thought to be distinct from declarative or episodic learning, which requires the hippocampus and medial temporal lobe structures (*Mishkin et al., 1984*; *Morris et al., 1986*; *Milner et al., 1998*; *Eldridge*

*et al., 2000*). On the other hand, work in both humans and rodents has also implicated the hippocampus in reward processing and motivated behavior, though the underlying mechanisms remain unclear (*Adcock et al., 2006*; *Gauthier and Tank, 2018*).

The hippocampus is also a target of dopaminergic projections. Dopamine receptors are expressed in the hippocampus, and in mice D1-class receptor expression is common in the dentate gyrus (DG) region (*Gangarossa et al., 2012*; *Kempadoo et al., 2016*). However, these dopaminergic projections come from locus coeruleus (LC) (*Kempadoo et al., 2016*; *Takeuchi et al., 2016*; *Chowdhury et al., 2022*), rather than the major dopamine cell groups in the ventral tegmental area (VTA) and substantia nigra pars compacta (SNc), which supply dopamine to the basal ganglia (*Björklund and Dunnett, 2007*; *Ikemoto, 2007*). The functional role of the dopaminergic LC-DG projection remains obscure.

In this study, we examined the contribution of dopaminergic signaling in the DG to operant learning and behavior. We found that mice could learn to perform a new action (pressing a lever) for optogenetic activation of D1 + neurons in the DG. In addition, using both optogenetics and in vivo pharmacological manipulations, we found that activation of LC dopaminergic neurons that project to the DG can also support self-stimulation. This effect depended on the activation of D1-like receptors. Finally, using in vivo calcium imaging in appetitive operant conditioning with food rewards, we found that D1 + DG neurons were more related to the goal-directed actions than simply non-contingent reward presentation.

## Results

To understand the role of D1 + neurons in the hippocampus, we tested whether selective stimulation of these neurons can reinforce operant behavior using a self-stimulation paradigm. We injected either a Cre-dependent channelrhodopsin (AAV5-DIO-ChR2) or a fluorescent control (DIO-eYFP) into D1-Cre mice (D1::ChR2$^{DG}$ or D1::eYFP$^{DG}$), producing selective expression of the excitatory opsin in D1 + neurons in the dentate gyrus (*Figure 1A–B*). Mice received photo-stimulation (500 ms, 20 Hz, 15 ms pulse width) following lever pressing on a fixed ratio schedule of reinforcement (*Figure 1C*). All D1::ChR2$^{DG}$ mice learned to press a lever for stimulation, whereas control mice did not (*Figure 1D*). These results suggest that D1::ChR2$^{DG}$ stimulation is sufficient to reinforce lever pressing. Interestingly, this form of self-stimulation is remarkably resistant to extinction, persisting after 8 days without any photostimulation.

Next, using retrograde tracing methods, we were able to map projections to the DG (*Figure 2A–D* & *Figure 2—figure supplement 1*). We confirmed significant LC projections to the DG, but we did not find significant VTA or SNc projections (*Figure 2E and H* & *Table 1*). Retrograde labeling of DG-projecting LC neurons is colocalized with tyrosine hydroxylase (TH), a marker for catecholamine neurons (e.g. dopamine, norepinephrine; *Figure 2F–H*). In contrast, there was no labeling in the VTA (*Figure 2H*, *Figure 2—figure supplement 1*). This finding suggests that the DG receives TH + projections from the LC rather than VTA.

We then tested whether the LC-DG projection is responsible for the self-stimulation effect observed. In order to manipulate the LC-DG pathway selectively, we injected AAV-Retro2-Cre into the DG and a Cre-dependent ChR2 (AAV5-DIO-ChR2) into the LC (*Figure 3A–B*). We found that ChR2$^{DG-LC}$ (n=8) mice also showed self-stimulation that is comparable to the stimulation of D1::ChR2$^{DG}$ neurons (*Figure 3C*).

The LC-DG projection releases both norepinephrine and dopamine (*Kempadoo et al., 2016*; *Takeuchi et al., 2016*). It is unclear which transmitter is responsible for the self-stimulation effect, though our observation on D1 + DG neurons (*Figure 1*) suggests that dopamine might be responsible. Consequently, to determine which of these transmitters is responsible for the observed effects, we used pharmacological manipulations in combination with pathway-specific optogenetic manipulations using the same self-stimulation paradigm (*Figure 3E* and *Figure 4*). Mice (n=8) trained on self-stimulation were tested after either receiving systemic injections of a β-adrenoceptor antagonist (propranolol), or a D1-antagonist (SCH 23390). β-adrenoceptor blockade did not produce any significant effects (*Figure 3D and F*). In contrast, D1-antagonist significantly impaired self-stimulation (*Figure 3E*).

To activate DG-projecting LC neurons robustly, we targeted the cell bodies of the DG-LC projections. But the LC has broad projections to many brain areas (*Schwarz and Luo, 2015*). To verify that our self-stimulation effects were not due to the activation of LC collaterals in other regions, we

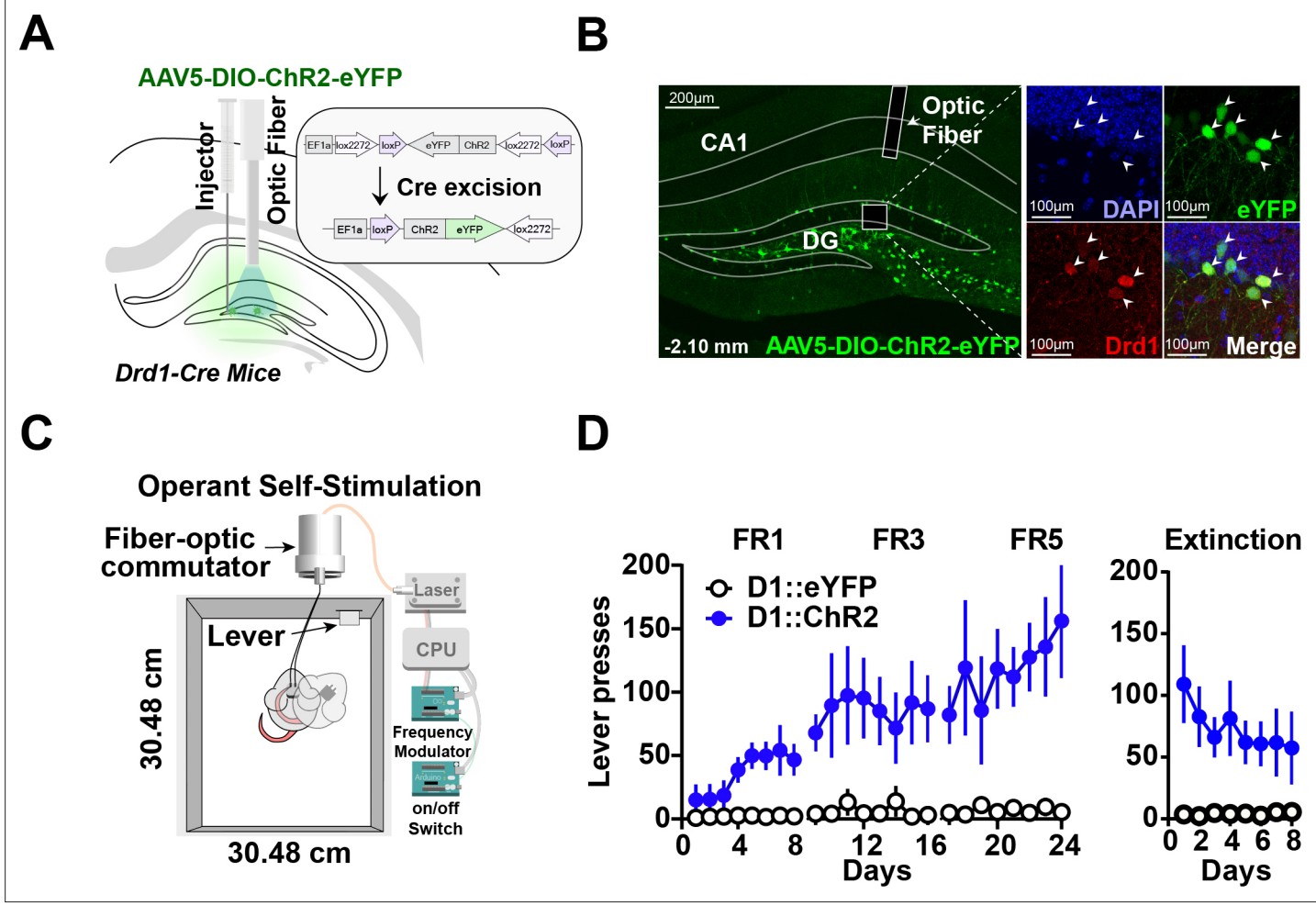

**Figure 1.** Optogenetic stimulation of D1 + neurons in the dentate gyrus is sufficient for operant self-stimulation. (**A**) Schematic of optic fiber placement above virally infected D1 + dentate gyrus (DG) neurons in D1-Cre mice. (**B**) *Left*, coronal section showing ChR2 expression in the DG. *Right*, magnified view of AAV infected DG neurons from inset colocalized with D1 receptors. White arrows indicate cell bodies. (**C**) Schematic of the operant self-stimulation chamber. (**D**) Lever pressing rate across three fixed ratios (FR1, FR3, FR5) schedules of reinforcement, and extinction (8 days each) for D1:Chr2-DG animals (n=8) and eYFP (n=8) controls. D1::Chr2-DG mice self-stimulated significantly more than controls (Two-way RM ANOVA, Group [ChR2 or eYFP] × Day, main effect of group $F_{(1,14)}$ = 16.59, p=0.0011, main effect of Day, $F_{(23, 322)}$ = 2.958, p=0.0078, and a significant interaction between day × group $F_{(23,322)}$ = 2.479, p=0.0003). During extinction, there was a significant main effect of group: $F_{(1,112)}$ = 58.87, p<0.0001, no significant effect of Day: $F_{(7, 112)}$ = 0.4571, p=0.8635, and no interaction: $F_{(7, 112)}$ = 0.8243, p=0.8243. Means +/−SEM for all graphs. DG, dentate gyrus; LC, Locus Coeruleus; scp, superior cerebellar peduncle; DAPI, 4′,6-diamidino-2-phenylindole. ****p<0.0001.

The online version of this article includes the following source data for figure 1:

**Source data 1.** The press rate (presses/min) of D1::ChR2 and D1::eYFP mice across FR1, FR3, and FR5 sessions.

**Source data 2.** The press rate (presses/min) of D1::ChR2 and D1::eYFP mice across extinction sessions.

performed local infusions of antagonists (*Figure 4A–C*, n=8). Infusions of a D1-antagonist into the DG significantly impaired self-stimulation, whereas propranolol showed no significant group differences in self-stimulation (*Figure 4D–E*). These results suggest that the reinforcing effects of LC-DG stimulation are due to the activation of D1 receptors by dopamine, rather than by norepinephrine.

Based on our self-stimulation results, we hypothesized that DG D1 + neurons may be preferentially activated during operant conditioning in general, including during actions that result in natural rewards rather than simply optogenetic stimulation of the LC-DG pathway. To test this, we performed in vivo calcium imaging of DG D1 + neurons while performing an operant lever pressing task with a food reward. We implanted a gradient index lens above the DG in D1-Cre mice (n=5) and injected them with a Cre-dependent calcium indicator (AAV9-syn-FLEX-jGCamp7f) (*Figure 5A–B*).

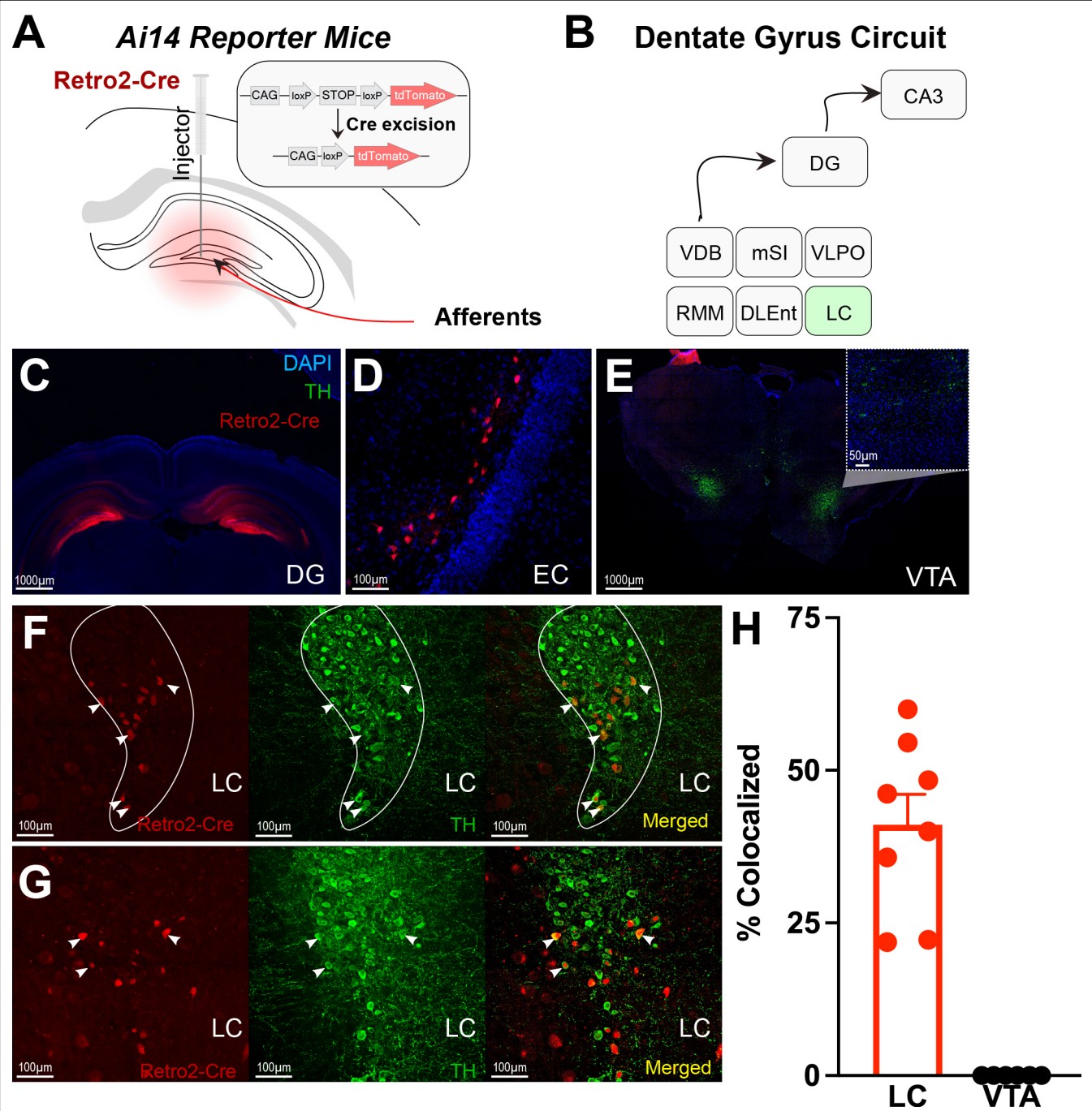

**Figure 2.** Retro-Cre tracing shows that the main catecholamine input to the dentate gyrus (DG) is the locus coeruleus (LC), not the ventral tegmental area. (**A**) Schematic of Retro2-Cre injection into the dentate gyrus in Ai-14 reporter mice. (**B**) Schematic summarizing all brain regions that project to the DG. Only LC is TH+. Abbreviations – Vertical diagonal band (VDB), Medial septal nucleus (MSN), Ventral lateral preoptic area (VLPO), retro mammillary bodies (RMM), dorsal lateral entorhinal cortex (DLEnt) (**C**) Injection site of the Retro2 showing the Cre-positive neurons. (**D–G**) Retrograde labeling of neurons in canonical brain regions that project to the hippocampus. (**D**) Entorhinal cortex (EC). (**E**) Limited retrograde labeling of neurons in the VTA, colocalized with tyrosine hydroxylase (TH). (**F & G**) Retrograde labeling of LC neurons in two out of four mice, colocalized with tyrosine hydroxylase. (**H**) Percent of colocalized neurons in the LC (n=8; four mice × two hemispheres) and VTA (n=6; three mice × two hemispheres). Unpaired *t*-test, p<0.0001. Mean and +/−SEM.

The online version of this article includes the following source data and figure supplement(s) for figure 2:

**Source data 1.** Percent of colocalized (TH+ and tdTomato+) neurons in the DG and VTA.

*Figure 2 continued on next page*

Figure 2 continued
**Figure supplement 1.** Retrograde tracing shows locus coeruleus (LC) projections, but not ventral tegmental area (VTA) projections to the dorsal dentate gyrus.

**Figure supplement 1—source data 1.** Percent of modulated D1 + DG neurons during different behavioral tasks.

We then recorded calcium transients from DG D1 + neurons during operant lever pressing for food rewards (*Figure 5A–C*) during lever pressing for food reward on fixed-ratio (FR) schedules (FR1, FR3, and FR5). We found distinct populations of DG D1 + neurons that were modulated by lever pressing. To see if the neural activity is action-contingent, we also used a control task in which pressing is not required. The reward was delivered non-contingently every 20 s, preceded by 1 s of white noise. On this task, there were far fewer significantly modulated DG D1 + neurons (n=6, 3% of the total population) compared to the operant task (*Figure 5F*, *Table 2*). To verify that the virus targets D1 + neurons in the DG, we quantified the percentage of neurons that are virally targeted that express D1 receptors. Using RNA scope, we found that GcAMP-7f was colocalized with D1 receptors (*Figure 6*).

To determine if the activity of these neurons reflected the spatial locations of the lever pressing or the action of the lever pressing itself, we used a discrete trial design with two levers (*Figure 7*). On each trial, one of the two levers was randomly selected to extend into the operant box. Once pressed, the lever would retract. The reward would then be delivered 1 s later. This task allowed us to compare the neural activity modulated by lever pressing and reward, as well as determine the spatial tuning of the same neurons. We found that several populations of dentate D1 + neurons (n=40, 16.5% of the total population) that were significantly modulated at the time of lever pressing (*Figure 7C*). One small population was modulated by reward delivery (n=14, 4.9% of the total population). Importantly, another population with significantly more neurons was responsive to lever pressing at either lever location (*Figure 7C–D*; n=22, 9.79% of the total population). These neurons were not spatially selective, as they were responsive when the lever was presented at different locations. However, we did find a small population that responded to only a single lever (*Figure 7C*; left lever: n=15, right lever: n=12; 4.2% of total population).

It is difficult to assess the spatial activity of neurons in operant tasks, as animals do not cover the arena equally but instead preferentially occupy specific task-relevant locations (*Figure 7—figure supplement 1*). To examine the stability of spatially related activity during operant conditioning, we used an FR5 task with two levers (*Figure 7—figure supplement 2*). We then used the same methods as described in *Skaggs et al., 1993* to identify spatially modulated neurons, and split the sessions into periods when the left or right lever was active. This allowed us to recalculate the center of mass of our identified spatial firing fields with two different lever locations. As mice mostly stayed close to the wall where the two levers and food port were located, we limited our analysis to the x-dimension

**Table 1.** Colocalization of tyrosine hydroxylase with retrograde dentate gyrus (DG) labeling in the locus coeruleus (LC).
Colocalization of tyrosine hydroxylase (TH) and Retro-cre labeling (n=8; four mice, two hemispheres), showing that at least some of the LC-DG neurons are TH positive. In contrast, no colocalization was found with retro-Cre and TH labeling in the VTA (n=6; three mice, two hemispheres), and ventral tegmental area (VTA) slices were not taken from one animal.

| Animal | Retro-cre label VTA (from DG) | Retro-Cre label LC (from DG) | LC colocalized with TH |
|---|---|---|---|
| Mouse 1 (LH) | 0 | 18 | 4 |
| Mouse 1 (RH) | 0 | 32 | 7 |
| Mouse 2 (LH) | 0 | 5 | 2 |
| Mouse 2 (RH) | 0 | 13 | 6 |
| Mouse 3 (LH) | 0 | 20 | 12 |
| Mouse 3 (RH) | 0 | 31 | 15 |
| Mouse 4 (LH) | n/a | 22 | 12 |
| Mouse 4 (RH) | n/a | 14 | 5 |

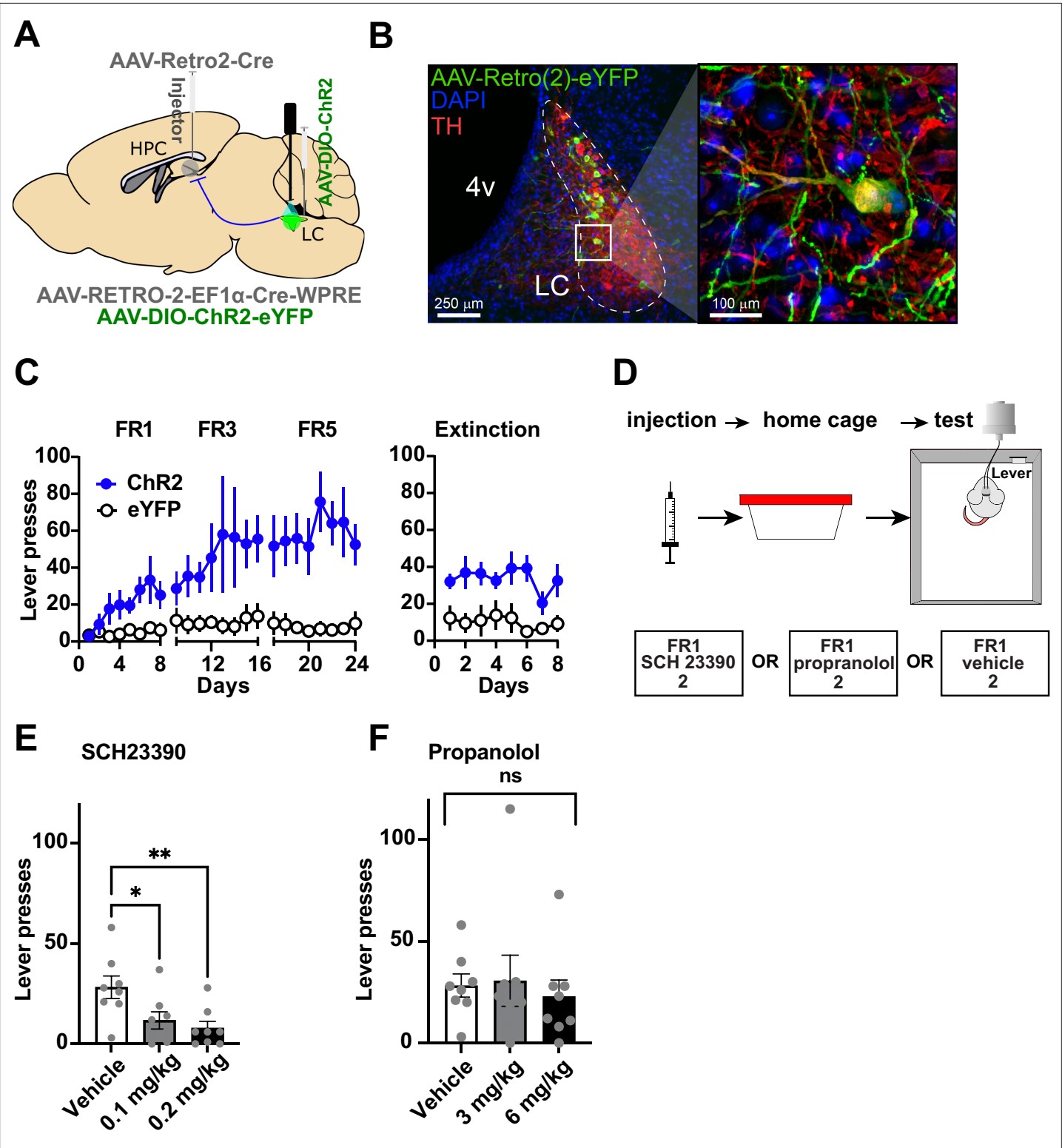

**Figure 3.** Locus coeruleus (LC) projections to the dentate gyrus (DG) contribute to operant learning. (**A**) Schematic showing selective targeting of LC-DG projection. (**B**) Left: Representative coronal section showing Cre-dependent ChR2 expression in the LC. Right: Magnified view from inset showing eYFP colocalization with tyrosine hydroxylase (TH) LC neurons. (**C**) Lever presses per session for three fixed ratios (FR1, FR3, FR5) schedules of reinforcement, and extinction (no stimulation following lever pressing) for Retro::ChR2 DG-LC animals (n=8) and eYFP (n=8) controls. Mice with ChR2 expressed in LC self-stimulated significantly more than controls (Two-way RM ANOVA Group [ChR2 or eYFP] × Day, the significant effect of group: $F_{(1,14)} = 13.09$, p=0.0031), the significant effect of day, $F_{(23, 322)} = 3.601$, p<0.0001; interaction of day × group: $F_{(23,322)} = 2.712$, p<0.0001. (**D**) Left,

*Figure 3 continued on next page*

*Figure 3 continued*

design of pharmacological experiments. After FR1 training, mice received IP injections of antagonists for either D1 (SCH 23390), or NE-beta receptors (propranolol). (**E**) D1-antagonist SCH 23390 significantly reduced self-stimulation of DG-projecting LC neurons: F(2, 14) = 6.9, p=0.008. Post hoc analysis (Dunnett's) shows that both doses reduced lever pressing relative to controls (0.1 mg/kg, p=0.02; 0.2 mg/kg, p=0.007). (**F**) The NE antagonist propranolol did not have any effect: F(2, 14) = 0.290, p=0.753. Means +/−SEM. HPC, hippocampus; 4 v, fourth ventricle. *p<0.05, **p<0.01.

The online version of this article includes the following source data for figure 3:

**Source data 1.** The press rate (presses/min) of LC-DG::ChR2 LC-DG::eYFP mice during extinction days.

**Source data 2.** The press rate (presses/min) of LC-DG::ChR2 LC-DG::eYFP mice across FR1, FR3, and FR5 sessions.

**Source data 3.** Total presses during vehicle and propranolol I.P. administration (3mg/kg and 6mg/kg).

**Source data 4.** Total presses during vehicle and SCH-23390 I.P. administration (0.1mg/kg and 0.2mg/kg).

which explained most of the variance in the neural activity. We found that the place field centers were significantly different when the lever is available. While we did find that occupancy varied in this switch task, the occupancy across other tasks was consistent, suggesting that spatial modulation does not depend on the type of task (e.g. operant vs Pavlovian) (*Figure 7—figure supplement 1*). In contrast, task-related neural activity in the DG depends on whether the task is action-contingent.

## Discussion

Together our results provide the first evidence that DG D1 + neurons may play a role in operant reinforcement. Using a cell-type-specific approach, we showed that activation of D1 + neurons in the DG is sufficient for self-stimulation (*Gangarossa et al., 2012*). Furthermore, our retrograde tracing identified that the DG receives TH + input from the LC and not from the VTA or SNc, suggesting that the LC is a source of dopaminergic projections to the DG. We found that mice will press a lever for optogenetic stimulation of the LC-DG projection. Blockade of D1 receptors, but not noradrenergic beta receptors, attenuated the self-stimulation of hippocampal-projecting LC neurons.

These findings build upon previous research that suggests that the LC supplies the primary dopaminergic input to the dorsal hippocampus (*Kempadoo et al., 2016*; *Takeuchi et al., 2016*). While previous work focuses on the LC-CA1 pathway, we examine a population of D1 + neurons in the DG that plays a role in operant reinforcement.

Both the hippocampus and the LC are known to be effective sites for intracranial self-stimulation (*Ursin et al., 1966*; *Crow et al., 1972*; *Ritter and Stein, 1973*). However, as non-selective electrical stimulation was used in classic studies, the precise circuit mechanisms underlying these observations remain unclear. In the present study, we have investigated a pathway for operant reinforcement that originates from LC neurons and targets D1 + DG neurons in the hippocampus. Future work will have to address whether this self-stimulation effect requires D1 + neurons or if it is a property of hippocampal neurons in general.

Self-stimulation behavior supported by stimulation of the LC-DG projections or the D1 + DG neurons appears to be different from that supported by stimulation of dopamine neurons in the VTA or SNc. First, the rate of lever pressing is much lower with LC-DG self-stimulation. Yet once established, the lever pressing was surprisingly resistant to extinction, persisting for many days after the termination of stimulation. This is very different from the self-stimulation of VTA or SNc DA pathways and their target regions; such classic self-stimulation behavior is rapidly extinguished in the absence of stimulation (*Gallistel, 1964*; *Olds, 1977*). Thus, classic self-stimulation supported by SNc and VTA DA neurons, which project mainly to the frontal cortex and basal ganglia, has a much stronger immediate effect on performance but rarely produces persistent actions in the long-term without stimulation. The effect on performance can largely be explained by a repetition of the action commands recently generated, but without DA release in the BG this repetition effect decays. Self-stimulation of the LC-DG pathway, in contrast, is less robust but is supported by some long-lasting memory of the stimulation, in accordance with recent findings that the LC projections to the hippocampus play a major role in long-term contextual memory (*Chowdhury et al., 2022*).

Our calcium imaging results showed that D1 + neurons in the DG are preferentially active during goal-directed instrumental actions compared to passive reward delivery. These findings suggest that the DA projections to the hippocampus contribute to the reinforcement of specific instrumental

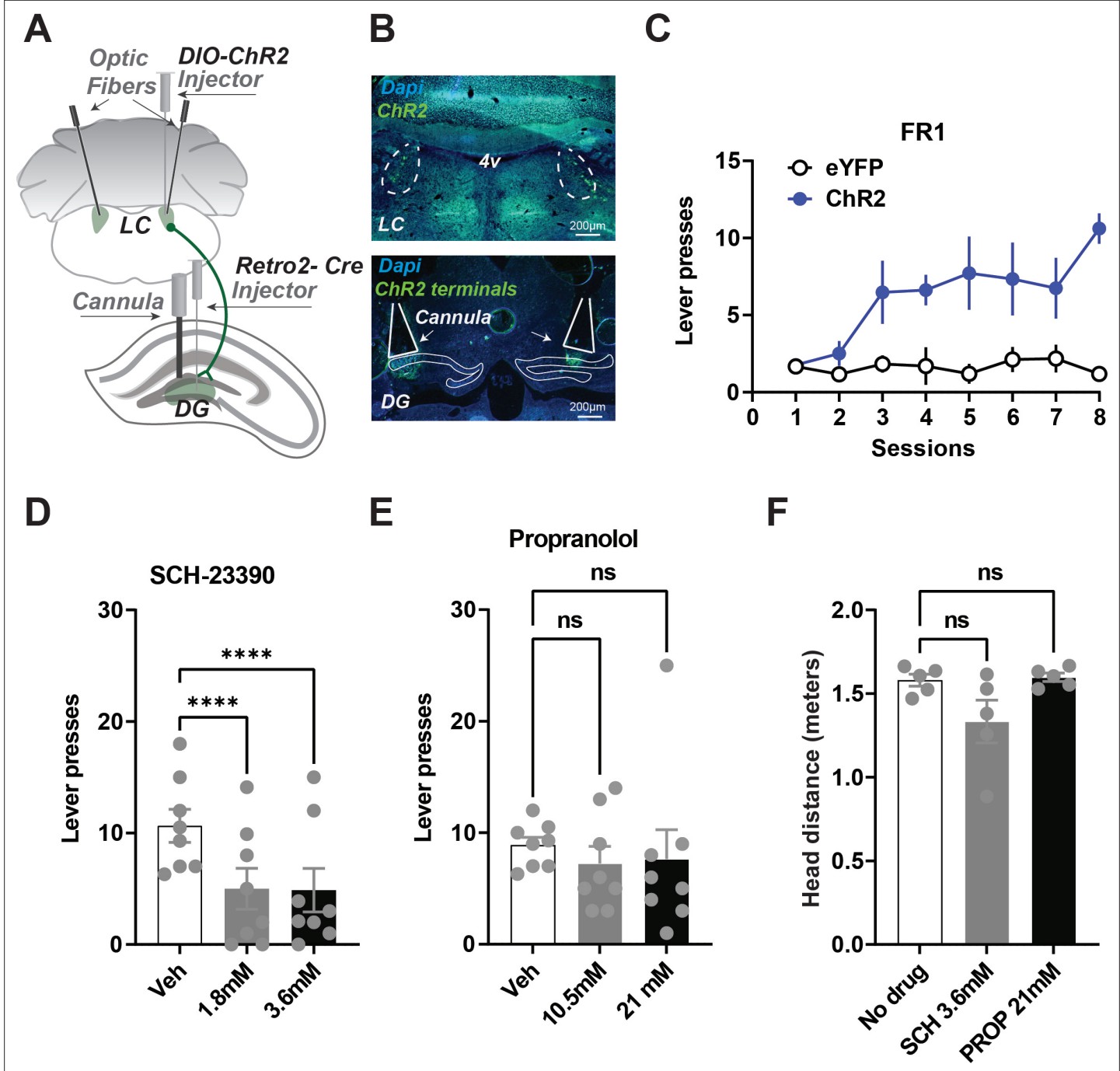

**Figure 4.** Local infusions of D1 but not NE beta antagonist into the DG reduces self-stimulation. (**A**) Schematic showing injection strategy for local drug infusions into the DG during self-stimulation of LC neurons that project to the DG (n=8). Cre expression was induced in LC neurons projecting to the DG by first injecting AAV-Retro-2 into the DG. An injection of a Cre-dependent virus (AAV5-DIO-ChR2-eYFP) was then made in the LC before optic fiber implantation. Canulae were used to inject DA and NE antagonists into the DG. (**B**) Top, representative coronal section showing ChR2 expression in the LC. Bottom, ChR2 terminals in DG and cannula tracks. (**C**) Acquisition of lever pressing Retro::ChR2 DG-LC mice or Retro::eYFP DG-LC (controls). Experimental animals self-stimulated more than controls (Two-way ANOVA [Day × Group], effect of Day $F_{(7, 84)} = 5.222$, $p<0.0001$; effect of group, $F_{(1,12)} = 37.98$, $p<0.0001$; interaction $F_{(7,84)} = 4.932$, $p<0.0001$). (**D**) One-way RM ANOVA showed D1 antagonist SCH-23390 significantly reduced lever pressing. There is a significant drug effect: $F_{(2,14)} = 39.16$, $p<0.0001$. Dunnett's multiple comparisons show both doses produced significant suppression of lever pressing (1.8 mM, $p<0.0001$; 3.6 mM, $p<0.0001$). (**E**) NE antagonist propranolol had no significant effect on self-stimulation. $F_{(2,14)} = 0.1912$, $p=0.8281$. Dunnett's multiple comparisons show no significant differences 10.5 mM, $p=0.7829$; 21 nM, $p=0.8624$. (**F**) Using DeepLabCut we tracked the distance traveled by the each animal and found no significant differences in the movement for the vehicle, SCH23390 (3.6 mM) or propranolol

*Figure 4 continued on next page*

Figure 4 continued

(21 nM) RM one-way ANOVA no effect of group, F(1, 4) = 3, p=0.1516. Means +/−SEM for all graphs. DG, dentate gyrus; LC, locus coeruleus; 4 v, fourth ventricle.

The online version of this article includes the following source data for figure 4:

**Source data 1.** Total presses during vehicle and propranolol administration through cranial cannula (21mM and 10.5mM).

**Source data 2.** Total presses during vehicle and SCH-23390 administration through cranial cannula (3.6mM and 1.8mM).

**Source data 3.** The press rate (presses/min) of LC-DG::ChR2 + DG cannula and LC-DG::eYFP + DG cannula mice across FR1 sessions.

**Source data 4.** Total head movement (meters) during SCH-23390(3.6mM) and propranolol(21mM) administration through cranial cannula.

actions. They are broadly in agreement with recent findings on entorhinal grid cells (*Butler et al., 2019*) and hippocampal CA1 cells (*Gauthier and Tank, 2018*), as well as LC terminals in CA1 that were preferentially active near a novel reward location (*Kaufman et al., 2020*). It remains to be determined how the activity of DG D1 + neurons changes during activation of LC neurons that project to the hippocampus, and if LC activation induces plasticity in the DG that is important for learning.

## Materials and methods

All experimental procedures were conducted in accordance with standard ethical guidelines and were approved by the Duke University Institutional Animal Care and Use Committee.

### Subjects

All behavioral data were collected from D1-cre mice (Cre targeted to *Drd1* locus, B6;129-Tg(Drd1-cre)120Mxu/Mmjax, Jackson Labs), and wild type (C57BL/6 J). Optogenetic control of D1-receptor expressing dentate hippocampal neurons was achieved with a double-floxed inverted recombinant AAV5 virus injection to express the excitatory opsin ChR2-eYFP. Viral infection in the dentate of the hippocampus was histologically verified with eYFP imaging colocalized against a D1 receptor antibody and DAPI staining. All mice were aged between 2–12 months old, and housed on a 12:12 light cycle, with tests occurring in the light phase. For calcium imaging experiments, mice were put on food restriction and maintained at 90% of their initial body weights.

### Viral constructs

CAV2-Cre was obtained from Institut de Génétique Moléculaire de Montpellier. rAAV5.EF1α.DIO.hChR2(H134R).eYFP, rAAV5.EF1α.DIO.eYFP, AAV9.hSyn.FLEX.jGCaMP7F, AAV(retro2).hSyn.EF1α.Cre.WPRE was obtained from the Duke University Vector Core. pAAV_hSyn1-SIO-stGtACR2-FusionRed was from Ofer Yizhar (Addgene viral prep # 105677-AAV1; https://n2t.net/addgene:105677; RRID:Addgene_105677).

### Pathway-specific retrograde tracing experiments

Retrograde anatomical tracing data was collected from Ai14 reporter mice (129S6-Gt(ROSA)26Sor^tm14(CAG-tdTomato)Hze/J, Jackson labs). Ai14 reporter mice have a loxP-flanked STOP cassette that is excised in the presence of Cre to promote transcription of a CAG promoter-driven red fluorescent protein variant (tdTomato). 50 nL of either Retro2-Cre (*Figure 2*), or canine adenovirus type 2 expressing Cre recombinase (CAV2-cre, *Figure 2—figure supplement 1*) was injected into the DG of Ai-14 reporter mice (AP: –2.0 mm relative to bregma, ML: ± 1.3 mm relative to bregma, DV: 2.0 mm from skull surface) (*Soudais et al., 2001*; *Tervo et al., 2016*).

### RNAscope

Three D1-Cre male mice were injected with the AAV9-hSyn-Flex-GCaMP7f in the DG (–2.0 & –2.2 AP; +\− 1.3 ML; 1.8 DV. from bregma). 3–4 weeks after injections, mice were euthanized with CO2, and their brain was quickly harvested and frozen in OCT for future RNAscope experiments using the Advanced Cell Diagnostics kit and probes (ACD). Brains were sectioned using a cryostat at a thickness of 20 micrometers and directly mounted on superfrost slides; the slides were then stored at –80.

On the experiment day, 4% PFA in PBS was chilled at 4 °C in a PFA-safe IHC container. Slides were removed from the –80 and immediately immersed in the pre-chilled PFA for 15 min at 4 °C. After

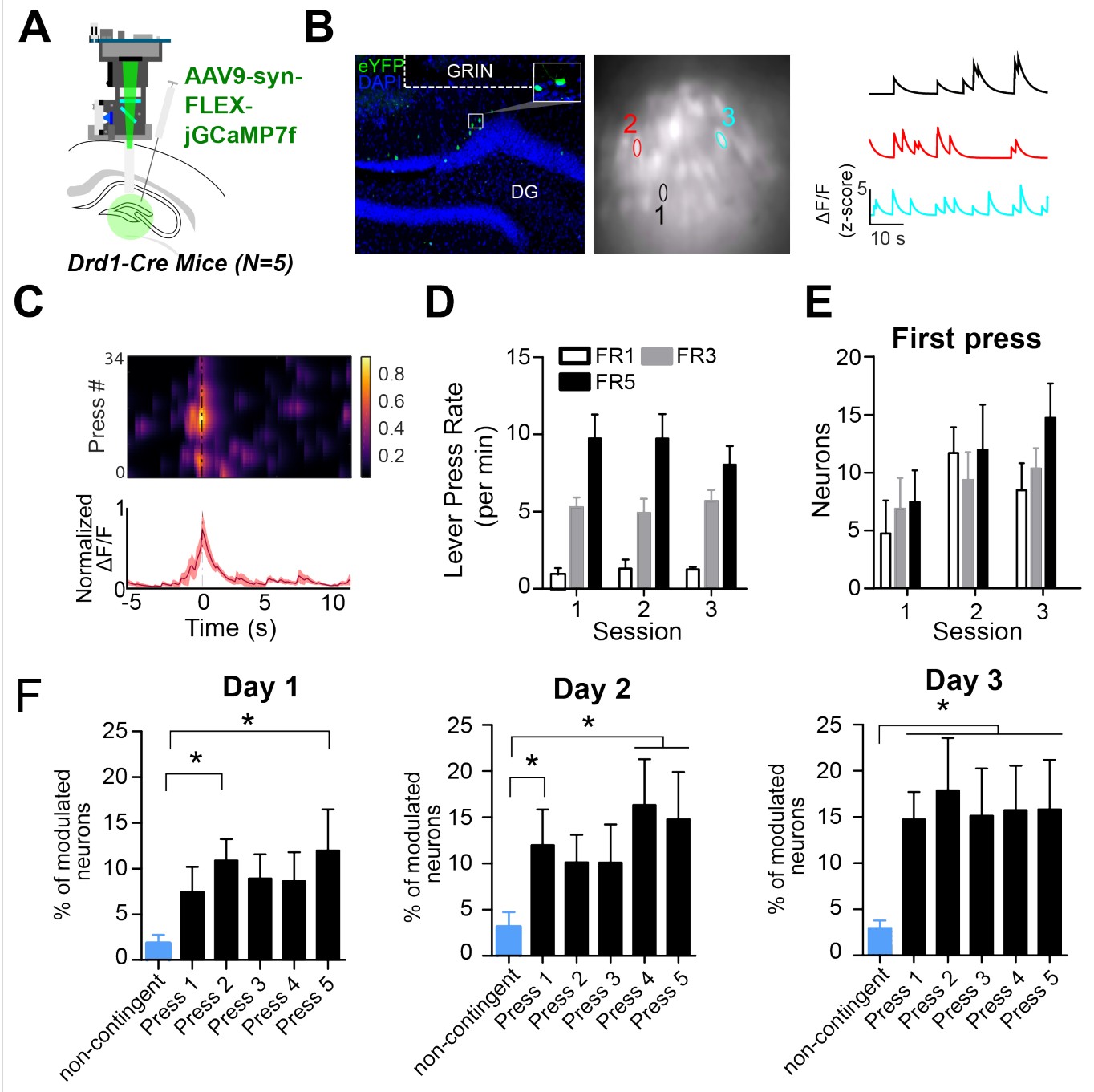

**Figure 5.** D1 + dentate gyrus neurons are significantly modulated by lever pressing during operant conditioning. (**A**) A schematic of a UCLA miniscope and GRIN lens implanted over jGCamp7f-infected D1 + cells in the dentate gyrus (DG) of D1-Cre mice for in vivo imaging (n=5). (**B**) *Left:* Representative coronal section showing GCaMP7f expression in D1 neurons of the DG. GRIN lens, marked in white. *Middle:* Imaging field of view with contours of identified neurons. *Right:* For example, calcium traces are significantly modulated by lever presses. (**C**) An example neuron showing increased calcium transient during lever pressing. *Top:* Heat map shows normalized calcium activity aligned to a single press as a function of time during fixed-ratio (FR) 1 trial. *Bottom:* Averaged calcium activity across all presses. (**D**) The press rates for FR1, FR3, and FR5 for each day of testing. The press rates increase across the FR schedule. (**E**) The percent of neurons modulated by the first press in each FR schedule across days. Two-way RM ANOVA FR schedule × Day, no effect of FR schedule $F_{(2,4)} = 0.6702$, p=0.5298; effect of day, $F_{(2,4)} = 4.536$, p=0.0213; no effect of interaction $F_{(2,4)} = 0.6403$, p=0.6389 (**F**) The percent of neurons modulated by a non-contingent reward task, or by each press in an FR5 task. Day 1, One-way RM ANOVA $F_{(4,5)} = 3.389$, p=0.0222. Day 2, One-way RM ANOVA $F_{(4,5)} = 16.19$, p=0.0046. Day 3, One-way RM ANOVA $F_{(4,5)} = 4.807$, p=0.0048. Dunnett's multiple comparison tests was used to compare the percent of press-modulated neurons to the non-contingent modulated neurons. Significance values are marked (*) for p<0.05.

The online version of this article includes the following source data for figure 5:

*Figure 5 continued on next page*

*Figure 5 continued*

**Source data 1.** Lever press rate (presses/min) during FR1, FR3 and FR5 sessions, number of modulated neurons during the first press, percent of modulated neurons during each press across days.

washing the slides with PBS, sections were dehydrated through a 5 min immersion at 50%, 70%, and 100% ethanol. Slides were then air-dried for 5 min at room temperature before incubating them for 30 min with protease IV (ACD #322336), then washed twice in PBS. Sections were then incubated with either the Drd1a probe (ACD #406491), the negative control probe (ACD #320871), or the positive control probe (ACD #320881) for 2 hr at 40 °C. Next, slides were washed twice using the wash buffer (ACD #310091) and then incubated for 30 min with Amp1, 15 min with Amp2, 30 min with Amp3, and finally 15 min with Amp4 Alt B-FL (ACD #320851). After the two washes, sections were then incubated with 5% Neutral Goat Serum (NGS) in 0.2% TBST for 1 hr at RT in the dark, then incubated for 1 hr with a primary antibody against GFP (1:1000; Millipore, AB16901) and for 2 hr with a secondary Alexa-fluorophore (488) conjugated antibodies (Invitrogen). Slides were mounted in Vectashield with DAPI (Vector Laboratories, CA) and a minimum of 5 images per mouse were acquired on an Olympus Fluoview confocal microscopy using a 60 X oil immersion objective.

Images were then processed using FIJI (https://imagej.net/Fiji/Download), and GFP + cells were identified and saved as individual ROIs. Using the GFP + cells, a mask was created to identify the presence of puncta (probe positive signals) within each ROI using the puncta analyzer plugin. The percentage of GFP + cells having puncta and the average number of puncta per GFP + cell were calculated in all three conditions. (*Figure 6*).

## Histology and immunohistochemistry

Mice were anesthetized and transcardially perfused with 0.1 M phosphate-buffered saline (PBS) followed by 4% paraformaldehyde (PFA) in order to confirm viral expression as well as optic fiber and GRIN lens placement. To confirm placement, brains were stored in 4% PFA with 30% sucrose for 72 hrs. Tissue was then post-fixed for 24 hr in 30% sucrose before cryostat sectioning (Leica CM1850) at 60 μm. Fiber and lens implantation sites were then verified.

To confirm eYFP expression in LC and DG neurons, sections were rinsed in 0.1 M PBS for 20 min before being placed in a PBS-based blocking solution. The solution contained 5% goat serum and 0.1% Triton X-100 and was allowed to sit at room temperature for 1 hr. Sections were then incubated with a primary antibody (polyclonal rabbit anti-TH 1:500 dilution, Thermo Fisher, catalog no. P21962;

**Table 2.** Calcium imaging neuron counts for each task.

| Task | Animal 1 | Animal 2 | Animal 3 | Animal 4 | Animal 5 |
|---|---|---|---|---|---|
| *Delay Task* | 64 | 62 | 22 | 59 | 100 |
| *FR1 Day 1* | 40 | 53 | 31 | 82 | 85 |
| *FR1 Day 2* | 42 | 48 | 26 | 99 | 45 |
| *FR1 Day 3* | 26 | 48 | 20 | 94 | 46 |
| *FR3 Day 1* | 26 | 54 | 23 | 90 | 32 |
| *FR3 Day 2* | 27 | 71 | 23 | 75 | 33 |
| *FR3 Day 3* | 27 | 75 | 27 | 80 | 33 |
| *FR5 Day 1* | 32 | 91 | 25 | 98 | 43 |
| *FR5 Day 2* | 28 | 106 | 27 | 100 | 57 |
| *FR5 Day 3* | 33 | 106 | 28 | 75 | 57 |
| *FR5 switch Day 1* | 99 | 65 | 18 | 56 | 63 |
| *FR5 switch Day 2* | 100 | 64 | 20 | 49 | 63 |
| *FR5 switch Day 3* | 96 | 66 | 21 | 54 | 62 |
| *Non-contingent* | 123 | 94 | 23 | 78 | 96 |

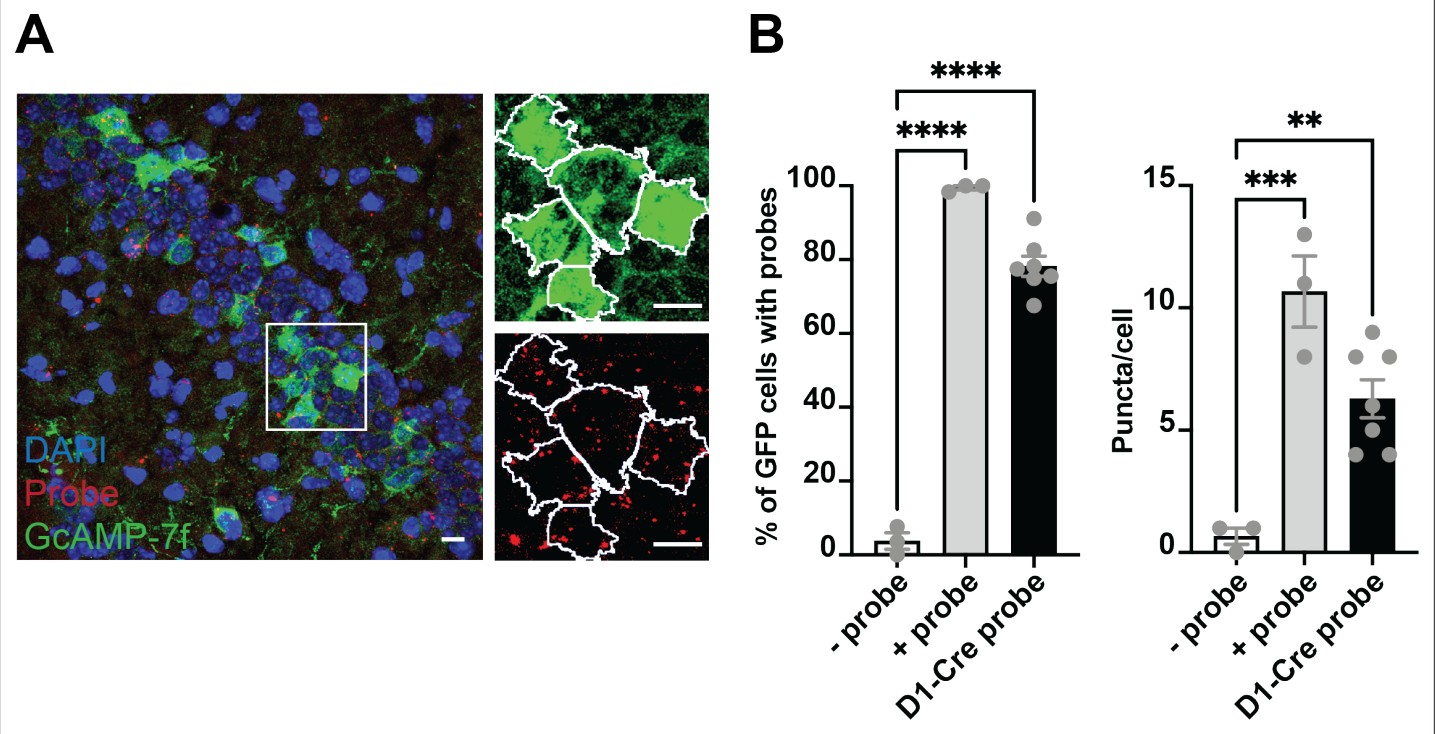

**Figure 6.** Colocalization of GcAMP-7f with D1 RNA scope probes. (**A**) *Left*, the representative coronal section from a Drd1a-cre mouse injected with GCaMP-7f. *Right*, zoomed-in view of green (GCaMP7f) and red (probe) channels, showing the outlines (white) of GFP + cells identified in FIJI. (**B**) *Left*, the percent of GFP cells colocalized with the negative control probe (n=3 sections; one brain, nine images), positive control probe (n=3 sections; one brain, five images), and Drd1a probe (n=7 sections; three brains, 38 images). One-way ANOVA showed a significant group difference (*F* = 229.2, p<0.0001). About 80% of GFP neurons colocalized with the Drd1a, much higher than the background negative control probe (Tukey's, p<0.0001). *Right*, there was also a significant group difference in puncta per cell (ANOVA, *F* = 19.51, p=0.0004) between Drd1a probe and negative control probe (Tukey's, mean diff = –5.619, adjusted p<0.0052) and positive control probe (Tukey's, mean diff = 4.381, adjusted p=0.0226). Thus the SIO/DIO constructs are only being expressed in D1 + neurons. Scale bars represent 10 um.

The online version of this article includes the following source data for figure 6:

**Source data 1.** Percent of GFP cells colocalized with negative control, positive control and Drd1a probes.

polyclonal chicken anti-EGFP, 1:500 dilution, Abcam, catalog no. ab13970) in blocking solution overnight at 4 °C. Sections were then rinsed in PBS for 20 min before being placed in a blocking solution with the secondary antibody used to visualize *TH* neurons in the LC (goat anti-rabbit Alexa Fluor 594, 1:1000 dilution, Abcam, catalog no. ab150080; goat anti-chicken Alexa Fluor 488, 1:1000 dilution, Life Technologies, catalog no. A11039) for 1 hr at room temperature. Sections were mounted and immediately coverslipped with Fluoromount G with DAPI medium (Electron Microscopy Sciences; catalog no. 17984–24). The placement was validated using an Axio Imager.V16 upright microscope (Zeiss) and fluorescent images were acquired and stitched using a Z780 inverted microscope (Zeiss).

## Co-localization analysis with tracing

In order to characterize projections to the dentate gyrus we injected 50 nL of AAV(retro2).hSyn. EF1α.Cre.WPRE into each hemisphere of the DG of Ai14 mice (four mice × two hemispheres, n=8; two females and two males). We then processed the slices and acquired images as described above. We opened the raw images taken from the Axio Imager V16 upright microscope (Zeiss) in Fiji to quantify the number of cells from a single coronal brain slice using eight-bit confocal images. A threshold was set to identify the neuronal cell bodies. The function 'fill holes' was then used to remove possible empty spaces within the selected cells. After converting the image to a mask, we ran the 'Analyze Particle' plug-in in Fiji to count the cells in each image. Using the Analyze Particle function, the masks taken were then counted to determine the number of co-localizing cells using the 'Colocalization Threshold' plug-in in Fiji.

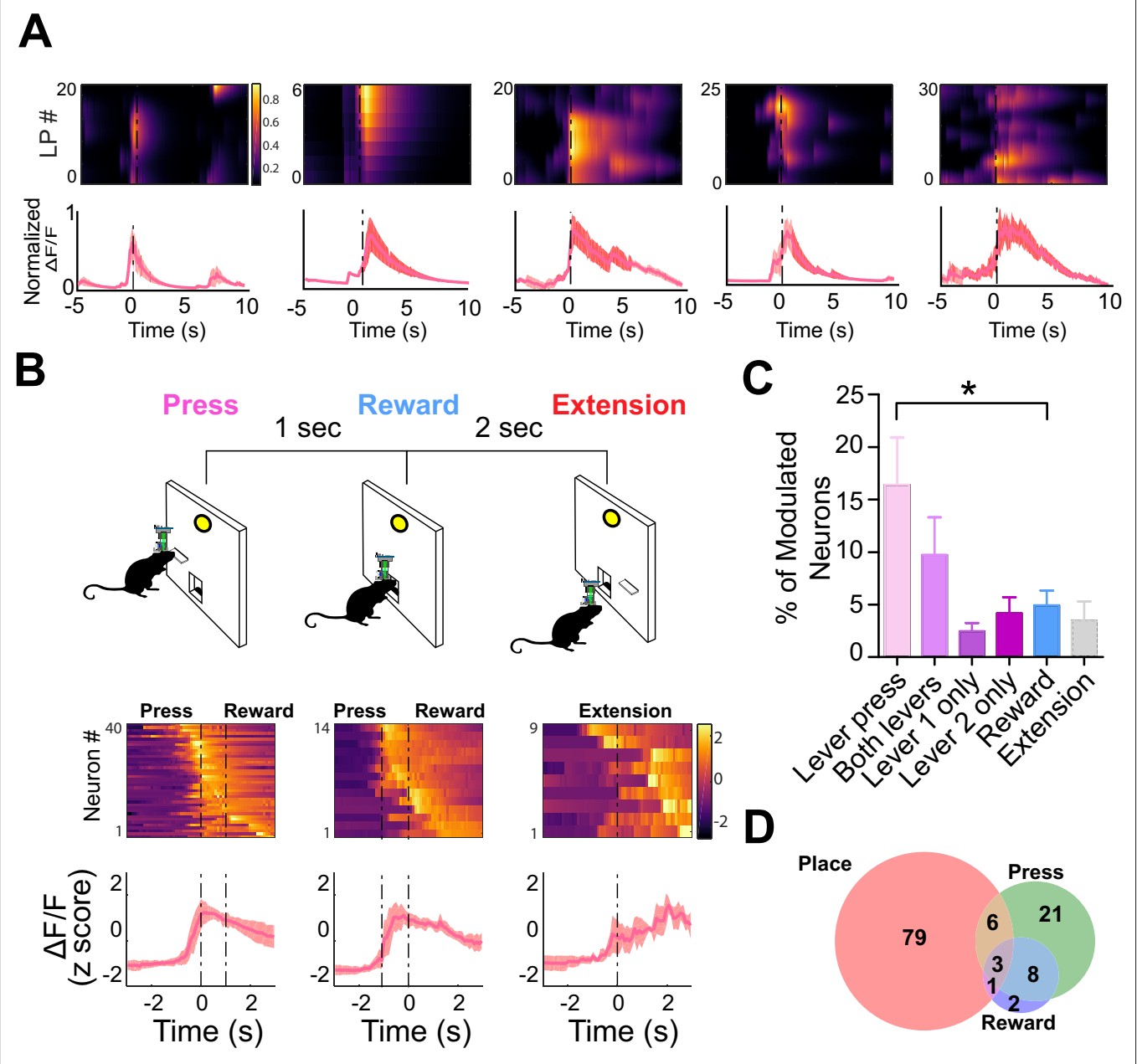

**Figure 7.** D1 + neurons in dentate gyrus (DG) are significantly modulated by lever pressing, but not by passive reward delivery. (**A**) One example neuron from each of the five calcium imaging animals, showed increased calcium transient during lever pressing. *Top*: Heat map shows normalized calcium activity aligned to a single press as a function of time during fixed-ratio (FR) 1 trial. *Bottom*: Averaged calcium activity across all presses. For the five different animals, we recorded n=64 (Animal 1), n=62 (Animal 2), n=22 (Animal 3), n=59 (Animal 4), and n=100 (Animal 5) (**B**) *Top:* Schematic of FR1 paradigm. Animals press one of two levers that are randomly presented, which then retracts followed by pellet delivery 1 s later. After 2 s, one of the two levers extends again at random. *Bottom*: Peri-event heat maps and average traces of calcium activity aligned to either *all lever presses*, *reward*, or *lever extension*. Only neurons that are significantly modulated around each event are shown. (**C**) Percentages of modulated neurons by each event. More neurons are modulated by lever pressing than reward delivery (One-way ANOVA, $F_{(5,24)}$ = 4.1077, p=0.0078; Dunnett's multiple comparisons: Lever press vs reward p<0.05). (**D**) Venn diagram displaying the number of neurons in a session modulated by spatial location (place), lever pressing, or reward delivery.

The online version of this article includes the following source data and figure supplement(s) for figure 7:

**Source data 1.** Percent of modulated neurons during lever press, both levers, left lever only, right lever only, reward, and extension.

**Figure supplement 1.** Comparison of neural activity in D1 + DG neurons during different behavioral tasks (**a**) Top-down views of the arena.

**Figure supplement 2.** Comparison of spatially modulated D1 + DG neurons while pressing two different levers.

## Optogenetic experiments

Mice were anesthetized with 2.0 to 2.5% isoflurane mixed with 1.0 L/min of oxygen for surgical procedures and placed into a stereotactic frame (David Kopf Instruments, Tujunga, CA). Meloxicam (2 mg/kg) and bupivacaine (0.20 mL) were administered prior to the incision. To optogenetically stimulate D1 + neurons in the hippocampus, adult D1-cre mice were randomly assigned to D1::ChR2(DG) (n=8, five males, three females, 8–10 weeks old) or D1::eYFP(DG) control groups (n=8, four males, four females, 8–10 weeks old). Craniotomies were made bilaterally above the hippocampus and AAV5-DIO-ChR2 was microinjected into the dentate gyrus through a pulled glass pipette (200 nL each hemisphere at 1 nL/s, AP: 2.0 mm relative to bregma, ML: ± 1.3 mm relative to bregma, DV: 2.0 mm from skull surface) using a microinjector (Nanoject 3000, Drummond Scientific). Optic fibers (SFLC230-10; 200 um core, 0.35 aperture, Ø1.25 mm, 6.4 mm Long SS Ferrule for MM Fiber, Ø231 µm Bore Size) were then implanted bilaterally above the dentate gyri. For pathway-specific experiments, wild-type mice were used to selectively target LC-Hipp (n=8, four males, four females, 8–10 weeks old) neurons by bilaterally injecting AAV(retro2).hSyn.EF1α.Cre.WPRE into the dentate gyrus (150 nL each hemisphere) combined with a Cre-dependent ChR2 virus injection into the locus coeruleus (AP: –5.45 mm relative to bregma, ML: ± 1.10 mm relative to bregma, DV: 3.65 mm from skull surface) before optic fiber placement (AP: –5.45 mm relative to bregma, ML: ± 1.10 mm relative to bregma, DV: 3.50 mm from skull surface). Controls received eYFP injections and fiber implants (n=8, four males, four females).

In addition, eight mice (four males, four females) were used in local infusion experiments in the DG. LC and DG surgeries were the same as pathway-specific manipulations, with the addition of cannulas (P1 technologies. AP: –2.00 mm relative to bregma, ML: ± 1.8 mm relative to bregma, DV: –1.4 mm, at a 10-degree angle). All optic fibers were secured in place with dental acrylic adhered to skull screws. Mice were group housed and allowed to recover for one week before experimentation.

## Operant self-stimulation

Standard operant boxes (Model ENV-007, MED Associates, Inc, Albans, VT) were housed in a light and sound-attenuating cubicles (Model ENV-019, MED Associates, Albans, VT). Each box is equipped with two levers. A Windows XP-based computer system running MED-PC Version IV Research Control & Data Acquisition System software (Med Associates, St. Albans, VT) is used to control the experimental equipment and record the data.

For self-stimulation experiments, a single lever was inserted at the start of the session. For each lever press animals received 500 ms of stimulation at 20 Hz (15 ms pulse width, 5 mW power). Animals were trained in 30 min sessions. Animals were tested for 32 consecutive days, and received eight sessions of FR1, FR3, and FR5, and then eight extinction sessions. To test for extinction, the lever was inserted but no stimulation was delivered for lever presses.

## Drug injections

Even low doses of DA antagonists are known to reduce self-stimulation, but low doses of NE antagonists have no effect on self-stimulation behavior (*Rolls et al., 1974*). In the present study, we selected doses that minimized effects on movement or arousal. The same mice used above were retrained after extinction (three FR1 sessions) to press a lever for stimulation of LC neurons that projected to the dentate gyrus. They were then tested with DA or NE antagonists (n=8; three males, five females). Mice alternated between testing days where they received an intraperitoneal injections (SCH23390 (Tocris) at 0.1 mg/kg and 0.2 mg/kg), propranolol (Sigma Aldrich) at 3 mg/kg or 6 mg/kg, or vehicle (phosphate-buffered saline), and training days with no injections. Each mouse had one testing day for each dose and drug combination (five total), and the order of the injections was determined pseudorandomly. Injections were given 30 min prior to the start of the session.

For local drug infusions with chronically implanted cannulae, we trained naive animals (n=5) for 8 days on self-stimulation and used the same experimental testing protocol for infusions of SCH23390 (1.8 mM or 3.6 m) or propranolol (10.5 mM 21 mM). The dose is determined based on previous work (*Takeuchi et al., 2016*). The drugs were infused at a rate of 0.0005 mL.

## Calcium imaging experiments

In order to target the dentate gyrus, AAV9-syn-FLEX-jGCamp7f was injected (50 nL) in four penetrations (A.P –2.0, M.L. 1.3, D.V. –2.2, –2.1, –2.0, –1.9) of D1-Cre mice (n=5, five males, 8–10 weeks old).

A gradient index (GRIN) lens (Inscopix: 1 mm × 4 mm, 1.8 mm DV) was then implanted over the injection site. Viral expression was checked three weeks post-injection, and a base plate was secured to the skull with dental cement. A UCLA miniscope was used to assess in vivo activity of D1 + neurons in the DG in freely moving animals. Images were collected from the miniscope using Bonsai (*Lopes et al., 2015*). This allowed for the simultaneous collection of calcium data with behavioral videos (Logitech c920). Calcium traces were then motion corrected (https://github.com/flatironinstitute/CalmAn; *Flatiron Institute, 2023*) and extracted using constrained nonnegative matrix factorization for calcium imaging data (https://github.com/zhoupc/CNMF_E; *Zhou, 2020*; *Zhou et al., 2018*). The extracted traces were then analyzed using custom MATLAB scripts. Imaging sessions lasted 8–10 min.

### FR training with a food reward (calcium imaging)

The behavioral tests used for calcium imaging were performed while the mice were food deprived to 85% of their free-feeding weight. There were six tasks used, with three days of testing for each task, for a total of 18 testing days. Each imaging session lasted approximately 10 min. Five mice were trained on a FR1 task where they received a pellet for pressing a lever. Subsequently, animals were moved to an FR3, and then FR5.

To examine the interaction of spatial location and lever pressing we used an *FR5 switch task* (*Figure 7—figure supplement 2*). In this task, a single lever into an operant chamber, and five responses resulted a pellet. Following five pellets (25 presses), the first lever is retracted, and a second lever is inserted. The mouse has to move to the other side of the food cup and press the other lever to earn a food reward.

In order to dissociate reward delivery and action production we used an FR1 schedule of reinforcement where the reward was delayed one second after the press (Two-lever FR1 delay task). We used two levers in this task. One lever is inserted on a given trial. If pressed the lever would retract, and a food pellet is delivered 1 s later. After a 2 s inter-trial interval, one of the two levers was randomly inserted. Following FR testing, mice received a reward at fixed intervals (20 s), preceded by 1 s of white noise (Non-contingent reward). The animals were not required to press a lever to receive the reward.

### Analysis

To assess significant increases in calcium activity, for each neuron we made a peri-event time histogram (PETH) between –3 s and 3 s aligned to relevant events (*da Silva et al., 2018*). Event times were considered between –500 ms and +500 ms around the relevant event. Baseline activity was considered –3000 to –1000 ms prior to the event. If the calcium activity was 99% above baseline for three consecutive 100ms bins then there was considered to be a significant increase in calcium activity. For tasks that involved two levers, we calculated the percent of neurons that were significantly modulated by either lever, and then excluded these neurons from our analysis of modulation by a specific lever. We identified spatially tuned cells by computing the spatial information contained in the calcium transients, compared with shuffled data (*Skaggs et al., 1993*). The data arena was split into 15 by 15 bins (2 × 2 cm each), and neurons were required to be active in active in at least three bins to be considered spatially modulated.

### Movement tracking (DeepLabCut)

To ensure the different compounds were not affecting gross movement we tracked the distance traveled during the vehicle, SCH23390, and propranolol infusions. To do this we used DeepLabCut (*Mathis et al., 2018*). For each video, 40 frames were labeled and the model was trained for 100,000 iterations. Outlier frames were extracted and relabeled and then the model was retrained. To ensure we had equal data sets 20 min of data was selected from each video.

### Acknowledgements

This work was supported by NIH grants DA040701 and NS094754 to HHY.

## Additional information

### Funding

| Funder | Grant reference number | Author |
|---|---|---|
| National Institute on Drug Abuse | DA040701 | Henry H Yin |
| National Institute of Neurological Disorders and Stroke | NS094754 | Henry H Yin |

The funders had no role in study design, data collection and interpretation, or the decision to submit the work for publication.

### Author contributions

Elijah A Petter, Conceptualization, Data curation, Formal analysis, Investigation, Methodology, Writing – original draft, Writing – review and editing; Isabella P Fallon, Formal analysis, Investigation, Methodology; Ryan N Hughes, Glenn DR Watson, Warren H Meck, Francesco Paolo Ulloa Severino, Investigation; Henry H Yin, Conceptualization, Resources, Formal analysis, Funding acquisition, Methodology, Writing – original draft, Project administration, Writing – review and editing

### Author ORCIDs

Isabella P Fallon (ID) http://orcid.org/0000-0002-1456-4152
Ryan N Hughes (ID) http://orcid.org/0000-0002-4999-0215
Francesco Paolo Ulloa Severino (ID) http://orcid.org/0000-0003-3725-9713
Henry H Yin (ID) http://orcid.org/0000-0003-1546-6850

### Ethics

All experimental procedures were conducted in accordance with standard ethical guidelines and were approved by the Duke University Institutional Animal Care and Use Committee (protocol number: 162-22-09).

### Decision letter and Author response

Decision letter https://doi.org/10.7554/eLife.83600.sa1
Author response https://doi.org/10.7554/eLife.83600.sa2

## Additional files

### Supplementary files

• MDAR checklist

### Data availability

All data generated or analysed during this study are included in the manuscript and supporting files.

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
