## [Editor Report]

These important findings indicate that dopamine signaling, arising from the locus coeruleus, and D1R expressing neurons in the dentate gyrus can support positive reinforcement. This is an exciting finding given the prior dearth of information on the role of dopamine signaling in the dentate gyrus. The evidence to support the claims is compelling. Rigorous optogenetic experiments, site specific pharmacology, tracing, and calcium imaging bring together a compelling argument that dopamine signaling in the dentate can play an important role in positive reinforcement. This manuscript will be of interest to those interested in dopamine, locus coeruleus and/or hippocampal function, learning or motivated behaviors.

---

## [Decision Letter]

**Decision letter after peer review:**

[Editors’ note: the authors submitted for reconsideration following the decision after peer review. What follows is the decision letter after the first round of review.]

Thank you for submitting the paper "Elucidating a locus coeruleus-hippocampal dopamine pathway for operant reinforcement" for consideration by *eLife*. Your article has been reviewed by 3 peer reviewers, and the evaluation has been overseen by a Reviewing Editor and a Senior Editor. The following individuals involved in review of your submission have agreed to reveal their identity: Kei Igarashi (Reviewer #2).

Comments to the Authors:

We are sorry to say that, after consultation with the reviewers, we have decided that this work will not be considered further for publication by *eLife*.

As you will see below, all the reviewers thought that this work contains significant findings that are of interest to the neuroscience community. In particular, the finding of the reinforcing effect of locus coeruleus (LC) neurons projecting to the dentate gyrus (DG) is potentially of great interest. However, the reviewers found several substantive issues that limit the validity of the conclusions. Specifically, the reviewers are concerned about a lack of experiments to establish the release of dopamine (or still potentially norepinephrine) from LC neurons projecting to DG. Currently, this conclusion was made using intraperitoneal injections of dopamine receptor antagonist. It is important to perform local pharmacological experiments for both norepinephrine and dopamine antagonists within the DG in self-stimulating mice to fully address this. Furthermore, many experiments reported in this paper, including experiments using norepinephrine and dopamine antagonists (Figure 3F), are underpowered. The number of animals (samples) needs to be increased throughout this work, in particular, the data supporting the specificity of dopamine as opposed to norepinephrine (Figure 3F). Another important concern raised by the reviewers is that it remains unclear whether there is something unique about putative D1R-expressing neurons in the DG for controlling instrumental actions.

Overall, the reviewers thought that the above issues are essential. Although these issues could be addressed by additional experiments, it will likely take more than 2-3 months. Because of these reasons, we decided to reject this work, at least in the current form. If the authors can fully address the above issues and each of the reviewers individual concerns, we would be happy to consider a revised manuscript as a new submission.

*Reviewer #1 (Recommendations for the authors):*

This manuscript by Petter et al., examines the role of putative D1R expressing neurons in the dentate gyrus in regulating operant behaviors in mice. The authors show that mice will learn to press a lever to receive optogenetic stimulation of D1R-Cre targeted cells compared to eYFP control mice. Next, they show that the DG receives input from the LC as previously described. They then show that mice will self-stimulate for activation of LC neurons that project to the DG, and that optogenetic inhibition of D1-DG neurons reduces the amount of self-stimulation of LC-DG neurons. Finally, the authors use miniscope based calcium imaging to show that DG D1R expressing neurons show changes in their activity timelocked to lever pressing. Overall, the work builds upon previous studies to support a role of LC projections to the DC in instrumental learning and reinforcement. While many of the findings support these conclusions, there are a number of issues with the data and paper in its current form.

1) Targeting DG cells in the D1R-Cre mouse. The authors imply that somehow D1R expressing cells in the DG are critical for instrumental behavior. However, it is unclear whether there is robust co-localization with D1R expression and the virally targeted cells. One image is presented to suggest this, but the authors need to quantify the percent of cells that are virally targeted that express D1R vs. those that do not. This has been performed for striatal tissue, but I am not aware of studies that have done this for the DG.

2) The amount of nosepoking to receive optical stimulation in all of the self-stimulation experiments seems low, and it is somewhat difficult to really assess this as it is presented as a lever press rate. I would be helpful to present the data in the total number of nosepokes per session for all groups. Even if the level of self-stimulation is low, they do use the appropriate control group (eYFP expressing controls) show the behavior is different between the groups.

3) The tracing experiments, although simple and straightforward, suffer from low n's. 2 hemispheres from 2 mice are presented in figure 2 for example. It would be preferable if at least 3-4 biological replicates are included in each dataset. These should also include both male and female mice whenever possible.

4) The antagonist experiments presented in Figure 3 are consistent with their hypothesis that dopamine is mediating the self-stimulation effect. However, these drugs are given IP and there is no way to know whether they are actually acting in the DG vs. other areas that may also reduce reinforcement.

5) For the self-stimulation of LC neurons experiment (Figure 3) it is unclear why a non-specific viral strategy was employed. The authors could have easily targeted TH expressing neurons that project from the LC to the DG instead of only targeting neurons based on their projection target. At the very least, the authors should perform a careful quantification of the percentage of the LC cells that are targeted using the retroAAV that are TH^+^.

6) The presentation of the miniscope data could use some improvement. For example, how many cells for each animal were recorded? How many trials were analyzed per session? All of the image data looks processed, and it would be nice to see some raw data presented alongside the processed data. I was also struck by how short the recording sessions were (10 minutes). Why did the authors record for such a short period of time?

*Reviewer #2 (Recommendations for the authors):*

In this manuscript, Petter and colleagues identify the neuromodulatory pathway from the locus coeruleus to the dentate gyrus of the hippocampus. The manuscript has a novelty on focusing in this previously undescribed circuit. However, I have several major concerns that some of the claims are not justified by the results.

1. I have a concern on the propranolol experiment (Figure 3F). Although the authors state that they did not see significant effect with propranolol, there is a trend of decreases (as mentioned by the author). The sample number here looks like n=5, which would not be achieving a sufficient power. Thus, it may be the case that this circuit functions using both dopamine AND norepinephrine.

2. I have a difficulty in interpreting LC-stimulation + DG inhibition experiment (Figure 4), as this can be interpreted with many possibilities. Rather, LC axon inhibition would achieve simple conclusion about the involvement of this circuit.*Reviewer #3 (Recommendations for the authors):*

In this study, the authors show that stimulation of Drd1-expressing dentate gyrus (DG) neurons in the mammalian brain promotes operant reinforcement, thereby expanding the role of the hippocampus traditionally studied with respect to episodic/spatial memory to instrumental learning. This is a very interesting finding, supported by a substantial array of behavioural observations. Yet, I do have three sets of comments/questions regarding the selectivity of the main observations, and thus the strength of the related conclusions.

First, the authors conclude from their experiments that D1-expressing DG cells constitute a specific population of hippocampal neurons that supports operant reinforcement learning. But how cell-type selective is this behavioural contribution? Notably, can Drd1-non-expressing DG neurons also be involved in this behavioural effect? That is, can self-stimulation be achieved with other DG neuron types? Interestingly, many retro-Cre labelled LC neurons that project to DG are not expressing tyrosine hydroxylase (see Table 1). Likewise, is this behavioural effect selective to LC-to-DG inputs? That is, can this effect be obtained with any input targeting Drd1-expressing DG neurons (see Supplementary Figure 1)?

Second, while the data supporting the direct contribution of the LC-to-DG pathway in operant reinforcement is convincing (irrespective of whether or not such a contribution is selective), the dopaminergic identity of this pathway remains elusive. Throughout the manuscript the authors keep making this sort of strong statement: e.g., "surprisingly, these neurons receive dopaminergic projections from the locus coeruleus …" (e.g., in the Abstract). As far as I understand the results, neither the release of dopamine from, nor the dopaminergic identity of, the LC inputs to DG are demonstrated. This claim is central to the work but seems to rely on two indirect observations. First, DG-targeting LC neurons are immuno-positive for tyrosine hydroxylase. Indeed, because this marks catecholamine neurons and is thus expressed by both dopamine and norepinephrine neurons. Second, systemic injections of the β-adrenoceptor antagonist propranolol does not prevent self-stimulation while the D1-antagonist SCH 23390 does. But the direct blockade of the LC-DG pathway is not established in these pharmacological experiments. Should other pathways be recruited in this behaviour (e.g., involving VTA or SNC to other, non-hippocampal circuits), then the current interpretation of this pharmacological blockade would be misleading.

Finally, hippocampal activity is strongly coupled to animal's speed (and thus the corresponding network states that report active exploration versus immobility). The observation that D1+ DG neurons are modulated by lever pressing during operant conditioning is interesting. The inclusion of the passive reward delivery is a good control. But speed should be formally controlled for (analyses in Figure 5 and 6). This can be done in at least two ways: by using speed-matched time windows across the two reward delivery conditions (i.e., active lever press versus passive delivery), and by reporting the measure of DG activity (Δ F/F) as a function of speed at the active reward delivery.

[Editors’ note: further revisions were suggested prior to acceptance, as described below.]

Thank you for resubmitting your work entitled "Elucidating a locus coeruleus-hippocampal dopamine pathway for operant reinforcement" for further consideration by *eLife*. Your revised article has been evaluated by Kate Wassum (Senior Editor) and a Reviewing Editor.

The manuscript has been improved but there are some remaining issues that need to be addressed, as outlined below:

As you can see, we very much appreciated your thorough response to the prior review. The manuscript is much improved. However, there are some remaining concerns that need to be addressed. These are summarized for your convenience below but are described in more detail in the reviewers' comments. In your revision please provide a point x point response to each reviewer's comment.

– There are some concerns, especially regarding Figure 3, whether drug effects on behavior are specific to reinforcement and not confounded by general effects on motor behavior. We feel this could be ideally addressed experimentally, though there may be other avenues to alleviate this concern.

– There are concerns that the conclusions that the reinforcement effects are mediated by dopamine and not norepinephrine hinge on the strong conclusions made about null results from experiments that could be underpowered (e.g., N=5) and also have high variability. As important conclusions rely on these findings, we think this concern should be addressed with additional experiments to increase power.

– Reviewer 3 makes a number of additional important points that we all agree should be addressed.

*Reviewer #1 (Recommendations for the authors):*

Most of my previous concerns are now resolved.

*Reviewer #2 (Recommendations for the authors):*

All of my previous concerns have been addressed.

*Reviewer #3 (Recommendations for the authors):*

1. There are concerns with some of the reinforcement data that clouds interpretation. The biggest concern I have is in Figure 3 where it is not possible to tell whether responding during the drug treatment challenges is truly being reinforced by LC-DG stimulation in these tests. These tests were run after 8d of extinction of a previous operant response, and there is no difference in responding between EXT day 8 and test day 1. In fact, they respond less during the testing. In figure 4, basal response rates are very low at the vehicle (10 responses/30min). Again, these rates are lower than extinction levels previously established. Are the authors sure that the drug effects on behavior are reinforcement-specific and not just general effects on motor behavior?

2. The only evidence that supports their claim that this is mediated by dopamine and not norepinephrine is the microinjection study, which is quite underpowered and also involves the question of reinforcement (above). This hinges on the strong conclusions made about null results from an underpowered set of experiments. For the authors are trying to rule out the contributions of NE, which is hard to do given n=5 where the mean effect size is a reduction of ~5-6 responses vs. ~2-3 responses on a task with fairly high variance to begin with. In particular, the 21nM propranolol group may be lower than the vehicle if the outlier with the increased responding is removed. This is important as the conclusion of the manuscript is that an LC-DG circuit mediates reinforcement via a dopamine input

3. It is not clear why in Figure 3 they didn't use a TH-specific viral strategy here. Since the manipulation in not cell-type specific, they really should have quantified these neurons. How many of the ChR2+ neurons are TH^+^?

4. The imaging studies were conducted on D1+ neurons, but not demonstrably related to LC inputs. With such small cell numbers activated here, and such a small TH^+^ population of LC neurons projecting to the DG – this is potentially a very small group of cells and the authors have not linked these responses to this specific input.

5. The discussion overstates a large number of the conclusions.

a. The authors point out in the text that it is interesting that this behavior doesn't really extinguish… yet this observation is left entirely unexplored. Perhaps, this circuitry is important for the acquisition of [reward-related] memories… but perhaps it is not critical to the maintenance of conditioned behavior. Some inhibition (or DA depletion) studies may give some insight into this possibility

b. Authors write: "These findings suggest that the LC supplies the primary dopaminergic input to the dorsal hippocampus (1), especially to a population of D1+ neurons in the DG (2), and that this dopaminergic pathway plays a critical role in operant reinforcement (3)."

i. It is not clearly demonstrated here that the LC provides dopaminergic input to the DG; however, it has been shown more clearly in previous studies. This language could be softened to suggest that they have data to support previous work.

ii. There is no direct evidence that LC projections modulate the D1+ population, so this may need to be softened as well.

iii. There is no evidence that this pathway is necessary, or even involved in natural reinforcement. This would require some circuit inhibition studies or pharmacological manipulations during sucrose reinforcement. It's also not clear what the authors believe the function of D1+ DG neurons to be.

c. Authors note "These findings suggest that the hippocampus contributes to the reinforcement of specific instrumental actions.". While this is possible, there is no clear evidence for this function – only evidence the calcium events were organized around lever presses. What about lever presses prior to learning? What about lever presses after extinction?

6. Is figure 1C showing a response isolated from an FR5 (as stated in the legend)? Were there no other responses within the 15-second timeframe of each response? How were these signals isolated and how was it ensured that the same data traces weren't being averaged several times but slightly offset from each other because of the temporal relationship between responses?

7. The response and the reinforcer delivery are only separated by 1 second… so it is challenging to distinguish whether there is a calcium response to reward (+1s) or not since fluorescence is already elevated at the time of reward consumption.

8. Similarly, the lever protracted 2sec after reward delivery – are the animals still consuming the sucrose pellet at this time?

9. Ultimately, it's unclear how they were able to categorize the neurons given the condensed task parameters and high variability exact spike time for the neurons shown in Figure 7B.

---

## [Author Response]

[Editors’ note: the authors resubmitted a revised version of the paper for consideration. What follows is the authors’ response to the first round of review.]

As you will see below, all the reviewers thought that this work contains significant findings that are of interest to the neuroscience community. In particular, the finding of the reinforcing effect of locus coeruleus (LC) neurons projecting to the dentate gyrus (DG) is potentially of great interest. However, the reviewers found several substantive issues that limit the validity of the conclusions. Specifically, the reviewers are concerned about a lack of experiments to establish the release of dopamine (or still potentially norepinephrine) from LC neurons projecting to DG. Currently, this conclusion was made using intraperitoneal injections of dopamine receptor antagonist. It is important to perform local pharmacological experiments for both norepinephrine and dopamine antagonists within the DG in self-stimulating mice to fully address this.

We have performed local pharmacological experiments using norepinephrine and dopamine antagonists within the DG (Figures 3-4). The results support our original conclusions.

Furthermore, many experiments reported in this paper, including experiments using norepinephrine and dopamine antagonists (Figure 3F), are underpowered. The number of animals (samples) needs to be increased throughout this work, in particular, the data supporting the specificity of dopamine as opposed to norepinephrine (Figure 3F).

We have also increased the number of animals for pharmacological experiments as well as tracing experiments.

Another important concern raised by the reviewers is that it remains unclear whether there is something unique about putative D1R-expressing neurons in the DG for controlling instrumental actions.

We have included discussion of this issue (see discussion) – “Future work will have to address whether this self-stimulation effect requires D1+ neurons or if it is a property of hippocampal neurons in general.”

Reviewer #1 (Recommendations for the authors):This manuscript by Petter et al., examines the role of putative D1R expressing neurons in the dentate gyrus in regulating operant behaviors in mice. The authors show that mice will learn to press a lever to receive optogenetic stimulation of D1R-Cre targeted cells compared to eYFP control mice. Next, they show that the DG receives input from the LC as previously described. They then show that mice will self-stimulate for activation of LC neurons that project to the DG, and that optogenetic inhibition of D1-DG neurons reduces the amount of self-stimulation of LC-DG neurons. Finally, the authors use miniscope based calcium imaging to show that DG D1R expressing neurons show changes in their activity timelocked to lever pressing. Overall, the work builds upon previous studies to support a role of LC projections to the DC in instrumental learning and reinforcement. While many of the findings support these conclusions, there are a number of issues with the data and paper in its current form.1) Targeting DG cells in the D1R-Cre mouse. The authors imply that somehow D1R expressing cells in the DG are critical for instrumental behavior. However, it is unclear whether there is robust co-localization with D1R expression and the virally targeted cells. One image is presented to suggest this, but the authors need to quantify the percent of cells that are virally targeted that express D1R vs. those that do not. This has been performed for striatal tissue, but I am not aware of studies that have done this for the DG.

We used RNAscope to quantify the number of virally target cells that colocalized with D1R expression. We found robust colocalization (Figure 6).

2) The amount of nosepoking to receive optical stimulation in all of the self-stimulation experiments seems low, and it is somewhat difficult to really assess this as it is presented as a lever press rate. I would be helpful to present the data in the total number of nosepokes per session for all groups. Even if the level of self-stimulation is low, they do use the appropriate control group (eYFP expressing controls) show the behavior is different between the groups.

Lever pressing was used. We now include total number of presses for all data.

3) The tracing experiments, although simple and straightforward, suffer from low n's. 2 hemispheres from 2 mice are presented in figure 2 for example. It would be preferable if at least 3-4 biological replicates are included in each dataset. These should also include both male and female mice whenever possible.

We added 2 mice with retro-Cre injected into the DG of Ai-14 mice (n = 4).

4) The antagonist experiments presented in Figure 3 are consistent with their hypothesis that dopamine is mediating the self-stimulation effect. However, these drugs are given IP and there is no way to know whether they are actually acting in the DG vs. other areas that may also reduce reinforcement.

We have performed the same experiments using local infusion of these drugs (Figure 4). Similar results were found.

5) For the self-stimulation of LC neurons experiment (Figure 3) it is unclear why a non-specific viral strategy was employed. The authors could have easily targeted TH expressing neurons that project from the LC to the DG instead of only targeting neurons based on their projection target. At the very least, the authors should perform a careful quantification of the percentage of the LC cells that are targeted using the retroAAV that are TH^+^.

We added quantification of the percentage of LC cells that are TH^+^ (Figure 2) and found that ~40% of the retro-cre labeled cells are also TH^+^.

6) The presentation of the miniscope data could use some improvement. For example, how many cells for each animal were recorded? How many trials were analyzed per session? All of the image data looks processed, and it would be nice to see some raw data presented alongside the processed data. I was also struck by how short the recording sessions were (10 minutes). Why did the authors record for such a short period of time?

We updated the Figure to reflect how many neurons were recorded per animal and included a table (Table 2) to show the number of neurons recorded from each animal for each task. We added more imaging data from other tasks that show the number of presses (“trials”) per session for each animal (Figure 6D, Figure 7A). Sessions were short because we wanted to avoid photobleaching.

Reviewer #2 (Recommendations for the authors):In this manuscript, Petter and colleagues identify the neuromodulatory pathway from the locus coeruleus to the dentate gyrus of the hippocampus. The manuscript has a novelty on focusing in this previously undescribed circuit. However, I have several major concerns that some of the claims are not justified by the results.1. I have a concern on the propranolol experiment (Figure 3F). Although the authors state that they did not see significant effect with propranolol, there is a trend of decreases (as mentioned by the author). The sample number here looks like n=5, which would not be achieving a sufficient power. Thus, it may be the case that this circuit functions using both dopamine AND norepinephrine.

We added animals to bring the sample number to N=8 for the systemic injections (Figure 3). We also performed local infusions with both NE and DA antagonists (cannulae in DG) to further confirm our effects.

2. I have a difficulty in interpreting LC-stimulation + DG inhibition experiment (Figure 4), as this can be interpreted with many possibilities. Rather, LC axon inhibition would achieve simple conclusion about the involvement of this circuit.

We removed this data, as it could be difficult to interpret.

Reviewer #3 (Recommendations for the authors):In this study, the authors show that stimulation of Drd1-expressing dentate gyrus (DG) neurons in the mammalian brain promotes operant reinforcement, thereby expanding the role of the hippocampus traditionally studied with respect to episodic/spatial memory to instrumental learning. This is a very interesting finding, supported by a substantial array of behavioural observations. Yet, I do have three sets of comments/questions regarding the selectivity of the main observations, and thus the strength of the related conclusions.First, the authors conclude from their experiments that D1-expressing DG cells constitute a specific population of hippocampal neurons that supports operant reinforcement learning. But how cell-type selective is this behavioural contribution? Notably, can Drd1-non-expressing DG neurons also be involved in this behavioural effect? That is, can self-stimulation be achieved with other DG neuron types? Interestingly, many retro-Cre labelled LC neurons that project to DG are not expressing tyrosine hydroxylase (see Table 1). Likewise, is this behavioural effect selective to LC-to-DG inputs? That is, can this effect be obtained with any input targeting Drd1-expressing DG neurons (see Supplementary Figure 1)?

We focused on the role of D1+ neurons and the DA projections from LC. We did not examine the role of other cell types in the DG. Future work will have to investigate how specific the effects are to D1 neurons. We found ~41% of the retrogradely labeled neurons colocalized with TH.

Second, while the data supporting the direct contribution of the LC-to-DG pathway in operant reinforcement is convincing (irrespective of whether or not such a contribution is selective), the dopaminergic identity of this pathway remains elusive. Throughout the manuscript the authors keep making this sort of strong statement: e.g., "surprisingly, these neurons receive dopaminergic projections from the locus coeruleus …" (e.g., in the Abstract). As far as I understand the results, neither the release of dopamine from, nor the dopaminergic identity of, the LC inputs to DG are demonstrated. This claim is central to the work but seems to rely on two indirect observations. First, DG-targeting LC neurons are immuno-positive for tyrosine hydroxylase. Indeed, because this marks catecholamine neurons and is thus expressed by both dopamine and norepinephrine neurons. Second, systemic injections of the β-adrenoceptor antagonist propranolol does not prevent self-stimulation while the D1-antagonist SCH 23390 does. But the direct blockade of the LC-DG pathway is not established in these pharmacological experiments. Should other pathways be recruited in this behaviour (e.g., involving VTA or SNC to other, non-hippocampal circuits), then the current interpretation of this pharmacological blockade would be misleading.

As mentioned above, we also included experiments using local infusion of antagonists to address this problem.

Finally, hippocampal activity is strongly coupled to animal's speed (and thus the corresponding network states that report active exploration versus immobility). The observation that D1+ DG neurons are modulated by lever pressing during operant conditioning is interesting. The inclusion of the passive reward delivery is a good control. But speed should be formally controlled for (analyses in Figure 5 and 6). This can be done in at least two ways: by using speed-matched time windows across the two reward delivery conditions (i.e., active lever press versus passive delivery), and by reporting the measure of DG activity (Δ F/F) as a function of speed at the active reward delivery.

In order to address these concerns, we report the measure of DG activity (df/f) as a function of speed during the reward delivery period. Specifically, we used a 4 second window (+/- 2 seconds) centered on reward delivery. For each trial and neuron pair, we take the average df/f in this 4 second window and plot it as a function of speed. In order to compare neurons on the same scale df/f was normalized by the maximum df/f for each neuron.

The data is plotted for all three tasks in Author response image 1. There is no significant speed modulation for the non-contingent task (R^2^=0.0591, p value: 0.2924) or the delay task (R^2^ = 0.0311 , p value: 0.4102). The FR5 task shows a weak negative relationship between speed and df/f (R^2^ = -0.094 , p value: 0.0031)

**Author response image 1. sa2fig1:** 

[Editors’ note: what follows is the authors’ response to the second round of review.]

The manuscript has been improved but there are some remaining issues that need to be addressed, as outlined below:As you can see, we very much appreciated your thorough response to the prior review. The manuscript is much improved. However, there are some remaining concerns that need to be addressed. These are summarized for your convenience below but are described in more detail in the reviewers' comments. In your revision please provide a point x point response to each reviewer's comment.– There are some concerns, especially regarding Figure 3, whether drug effects on behavior are specific to reinforcement and not confounded by general effects on motor behavior. We feel this could be ideally addressed experimentally, though there may be other avenues to alleviate this concern.

We analyzed total distance traveled for each drug condition and did not find effects on motor behavior (Figure 4F).

– There are concerns that the conclusions that the reinforcement effects are mediated by dopamine and not norepinephrine hinge on the strong conclusions made about null results from experiments that could be underpowered (e.g., N=5) and also have high variability. As important conclusions rely on these findings, we think this concern should be addressed with additional experiments to increase power.

We have performed additional experiments to increase N.

Reviewer #3 (Recommendations for the authors):1. There are concerns with some of the reinforcement data that clouds interpretation. The biggest concern I have is in Figure 3 where it is not possible to tell whether responding during the drug treatment challenges is truly being reinforced by LC-DG stimulation in these tests. These tests were run after 8d of extinction of a previous operant response, and there is no difference in responding between EXT day 8 and test day 1. In fact, they respond less during the testing. In figure 4, basal response rates are very low at the vehicle (10 responses/30min). Again, these rates are lower than extinction levels previously established. Are the authors sure that the drug effects on behavior are reinforcement-specific and not just general effects on motor behavior?

We believe that the drug effects on behavior are reinforcement specific. The animals used for LC-hippocampal pathway stimulation with intrahippocampal cannula were all naïve to start the experiments. These animals showed an increase in responding compared to controls (Figure 4C) and significant decreases in lever pressing in response to SCH23390 but not propranolol (Figure 4D,E). Further, we analyzed the movement of these animals using DeepLabCut and found that the head movement of these animals was unchanged by the drug infusions (Figure 4F). The change in lever pressing but not gross movement suggests these are likely reinforcement specific.

2. The only evidence that supports their claim that this is mediated by dopamine and not norepinephrine is the microinjection study, which is quite underpowered and also involves the question of reinforcement (above). This hinges on the strong conclusions made about null results from an underpowered set of experiments. For the authors are trying to rule out the contributions of NE, which is hard to do given n=5 where the mean effect size is a reduction of ~5-6 responses vs. ~2-3 responses on a task with fairly high variance to begin with. In particular, the 21nM propranolol group may be lower than the vehicle if the outlier with the increased responding is removed. This is important as the conclusion of the manuscript is that an LC-DG circuit mediates reinforcement via a dopamine input

We performed additional experiments and increased the N in each group to 8.

3. It is not clear why in Figure 3 they didn't use a TH-specific viral strategy here. Since the manipulation in not cell-type specific, they really should have quantified these neurons. How many of the ChR2+ neurons are TH^+^?

We did quantify the number of ChR2+ neurons that are TH^+^ in figure 2H.

4. The imaging studies were conducted on D1+ neurons, but not demonstrably related to LC inputs. With such small cell numbers activated here, and such a small TH^+^ population of LC neurons projecting to the DG – this is potentially a very small group of cells and the authors have not linked these responses to this specific input.

The imaging experiments merely attempted to show the properties of D1+ neurons in the DG during operant behavior. Whether these neurons receive TH^+^ LC projections cannot be determined, as doing pathway specific stimulation while imaging is not technically feasible. In the paper we did not make any specific claims about the relationship between TH^+^ LC projections and neurons we imaged. We do know from previous work that TH^+^ axons strongly innervate the hippocampus (Kempadoo et al., Kaufman et al.,2020, Takeuchi et al., 2016). While we have not linked the responses of the cells directly to LC activity, we find it probably that the LC modulates these cells based on the changes in behavior with D1-DG activation of LC-DG activation.

5. The discussion overstates a large number of the conclusions.a. The authors point out in the text that it is interesting that this behavior doesn't really extinguish… yet this observation is left entirely unexplored. Perhaps, this circuitry is important for the acquisition of [reward-related] memories… but perhaps it is not critical to the maintenance of conditioned behavior. Some inhibition (or DA depletion) studies may give some insight into this possibility

Testing reward-related memories is beyond the scope of these studies. We agree it would be interesting for future studies to inhibit D1-DG neurons during acquisition and extinction of conditioned behavior.

b. Authors write: "These findings suggest that the LC supplies the primary dopaminergic input to the dorsal hippocampus (1), especially to a population of D1+ neurons in the DG (2), and that this dopaminergic pathway plays a critical role in operant reinforcement (3)."i. It is not clearly demonstrated here that the LC provides dopaminergic input to the DG; however, it has been shown more clearly in previous studies. This language could be softened to suggest that they have data to support previous work.

We softened the language to reflect that this claim about DA inputs to the DG is shown by previous work, rather than our findings.

ii. There is no direct evidence that LC projections modulate the D1+ population, so this may need to be softened as well.

We have revised our discussion to point out the lack of direct evidence.

iii. There is no evidence that this pathway is necessary, or even involved in natural reinforcement. This would require some circuit inhibition studies or pharmacological manipulations during sucrose reinforcement. It's also not clear what the authors believe the function of D1+ DG neurons to be.

We have revised our discussion to point out the lack of direct evidence, but our calcium imaging data shows systematic changes in press-related activity during training.

c. Authors note "These findings suggest that the hippocampus contributes to the reinforcement of specific instrumental actions.". While this is possible, there is no clear evidence for this function – only evidence the calcium events were organized around lever presses. What about lever presses prior to learning? What about lever presses after extinction?

An additional piece of evidence supporting the role of these hippocampal neurons in operant conditioning is that there are more neurons responsive to lever pressing compared to noncontingent reward delivery. There are no lever presses prior to learning in such a task, since lever pressing is learned.

6. Is figure 1C showing a response isolated from an FR5 (as stated in the legend)? Were there no other responses within the 15-second timeframe of each response? How were these signals isolated and how was it ensured that the same data traces weren't being averaged several times but slightly offset from each other because of the temporal relationship between responses?

In figure 5C there was a typo and the example neuron was actually taken from an FR1 session.

We have updated the legend to reflect this. There are no other responses within the time frame.

7. The response and the reinforcer delivery are only separated by 1 second… so it is challenging to distinguish whether there is a calcium response to reward (+1s) or not since fluorescence is already elevated at the time of reward consumption.

This is partially addressed by the non-contingent reward task, where there is no operant conditioning component.

8. Similarly, the lever protracted 2sec after reward delivery – are the animals still consuming the sucrose pellet at this time?

It is possible that the animals are still consuming the sucrose pellet at this time. If they are still consuming the pellet they can wait to respond on the lever. The lever does not retract after a fixed amount of time, and therefore the animal can press it whenever it is ready.

9. Ultimately, it's unclear how they were able to categorize the neurons given the condensed task parameters and high variability exact spike time for the neurons shown in Figure 7B.

We used a non-contingent reward task so we could examine how many cells were responsive when no operant behavior was required for a reward. In this task we found significantly fewer modulated neurons compared to a task where lever pressing was required.